# Characterizing the role of early life factors in machine learning-based multimorbidity risk prediction

**Vien Ngoc Dang**[1]*, **Charlotte Cecil**[2,3,4], **Carmine M. Pariante**[5,6], **Jerónimo Hernández-González**[7], **Karim Lekadir**[1,8]

**1** Departament de Matemàtiques i Informàtica, Facultat de Matemàtiques i Informàtica, Universitat de Barcelona, Barcelona, Spain, **2** Department of Child and Adolescent Psychiatry/Psychology, Erasmus MC, University Medical Center Rotterdam, Rotterdam, The Netherlands, **3** Department of Epidemiology, Erasmus MC, Rotterdam, University Medical Center Rotterdam, Rotterdam, The Netherlands, **4** Molecular Epidemiology, Department of Biomedical Data Sciences, Leiden University Medical Center, Leiden, The Netherlands, **5** Institute of Psychiatry, Psychology and Neuroscience, Department of Psychological Medicine, Maurice Wohl Clinical Neuroscience Institute, King's College London, London, United Kingdom, **6** National Institute for Health Research (NIHR) Mental Health Biomedical Research Centre at South London and Maudsley NHS Foundation Trust, London, United Kingdom, **7** Departament d'Informàtica, Matemàtica Aplicada i Estadística, Universitat de Girona, Girona, Spain, **8** Institució Catalana de Recerca i Estudis Avançats (ICREA), Barcelona, Spain

* dangn@ub.edu

**Data availability statement:** Data from this study are from the UK Biobank (UKB), available upon approved application. Researchers can register and request access at https://www.ukbiobank.ac.uk/enable-your-research/register.

## Abstract

Recent evidence suggests that psycho-cardio-metabolic (PCM) multimorbidity finds its origins in exposure to early-life factors (ELFs), making the exploration of this association crucial for understanding and effective management of these complex health issues. Moreover, risk prediction models for cardiovascular diseases (CVD) and diabetes, as recommended by current clinical guidelines, typically demonstrate sub-optimal performance in clinically relevant sub-populations where these ELFs may play a substantial role. Our methodological approach investigates the contribution of ELFs to machine-learning-based risk prediction models for comorbid populations, incorporating a wide set of early-life and proximal variables, with a special focus on prenatal and postnatal ELFs. To address the complexity of integrating diverse early-life and proximal factors, we leverage models capable of handling high-dimensional, heterogeneous data sources to enhance prediction accuracy in complex clinical populations. The long-term predictive ability of ELFs, along with their influence on model decisions, is assessed with the learned models, and global and local model-agnostic interpretative techniques allow us to elucidate some interactions leading to multimorbidity. The data for this study is derived from the UK Biobank, showcasing both the strengths and limitations inherent in utilizing a single, large-scale database for such research. Our results show enhanced predictive performance for CVD (AUC-ROC: +7.9%, Acc: +14.7%, Cohen's d: 1.5) among individuals with concurrent mental health issues (depression or anxiety) and diabetes. Similarly, we demonstrate improved diabetes risk prediction (AUC-ROC: +12.3%, Acc: +13.5%, Cohen's d: 2.5) in those with concurrent mental health conditions and CVD. The inspection of these models, which integrate a large set of ELFs and other predictors (including the 7-core Framingham and UKDiabetes variables), provides key information that could

The results for UKB in this study will be returned to the UK Biobank within 6 months since publication, as required.

**Funding:** This study was supported by funding from the European Commission under Grant Agreement No. 101137146, STAGE. The funders had no role in study design, data collection and analysis, decision to publish, or preparation of the manuscript.

**Competing interests:** In the past three years, Professor Carmine Pariante has conducted research on inflammation and mental health in collaboration with Compass Pathways (UK) and the Wellcome Trust strategy award to the Neuroimmunology of Mood Disorders and Alzheimer's Disease (NIMA) Consortium (2015–2023; 104025/Z/14/Z), which was also funded by Janssen, GlaxoSmithKline, Lundbeck, and Pfizer. He is also funded by the NIHR Biomedical Research Centre at the South London and Maudsley NHS Trust and King's College London. The rest of the authors declare that no competing interests exist.

lead to a more profound understanding of psycho-cardio-metabolic multimorbidity. Our findings highlight the utility of incorporating life-course factors into risk models. Integrating a diverse range of physiological, psychological, and ELFs becomes particularly pertinent in the context of multimorbidity.

## Author summary

Many people experience multiple chronic diseases at the same time, such as heart disease, diabetes, and mental health conditions, which makes their health harder to predict and manage. This challenge contributes to health disparities, as individuals with multiple conditions often receive less accurate risk assessments and have limited access to tailored care. Research suggests that experiences early in life, from prenatal development to childhood, can shape long-term health, but these factors are rarely included in disease prediction models. In this study, we used machine learning to understand how ELFs contribute to predicting the risk of heart disease and diabetes, especially in people with mental health conditions. Our results show that including ELFs makes risk predictions more accurate, particularly for people affected by multiple diseases. We also visualize the influence of these factors in health risk prediction, helping to better understand the connection between early-life experiences and later disease. This research highlights the importance of considering life-course factors in disease prevention, paving the way for more personalized and fair approaches to healthcare.

## Introduction

The World Health Organisation classifies cardiovascular diseases (CVD) and diabetes among the foremost non-communicable diseases worldwide [1]. While CVD is the leading cause of global mortality [2], the prevalence of diabetes is rapidly increasing, imposing a significant burden on healthcare systems [3]. Apart from their distinct complexities, these conditions frequently intersect with mental health disorders, such as depression and anxiety [4], and also co-occur with one another [5], suggesting shared environmental and genetic factors that contribute to multimorbidity patterns [6]. However, the mechanisms leading to psycho-cardio-metabolic (PCM) multimorbidity, characterized by the simultaneous presence of two or more chronic conditions, are not well understood. Epidemiological studies have shown an association between early-life factors (ELFs), spanning prenatal, postnatal, and childhood periods, and the development of PCM conditions [7]. Prenatal factors, such as maternal smoking or challenges like poor fetal growth, as well as postnatal and childhood experiences like abuse or neglect, substantially influence long-term health outcomes, setting the stage for multiple chronic conditions [8]. These ELFs have been associated with an elevated risk of developing depression [9], anxiety [10], CVD [11], and diabetes [12] later in life, contributing individually to the risk profile of these disorders and underscoring the importance of considering these factors in understanding and addressing the patterns of multimorbidity.

The intricate nature of PCM multimorbidity, characterized by complex interrelationships among various health conditions and risk factors, poses substantial challenges in accurate disease prediction, particularly in the context of comorbid conditions. This brings us to another pivotal concern: the efficacy of risk prediction models currently recommended by clinical guidelines, which typically demonstrate sub-optimal performance, especially in clinically relevant sub-populations [13]. For example, in predicting CVD, the Framingham Risk Score,

as recommended by the 2010 American College of Cardiology/American Heart Association (ACC/AHA) guidelines [14], has been demonstrated to be less effective in individuals with diabetes [13]. This limitation is largely attributable to the restrictive assumptions of the model and limited predictors, which frequently presuppose a linear relationship between risk factors and disease outcomes [15]. Such simplifications may not fully capture the complex and often non-linear dynamics inherent in disease progression. For instance, associations between risk factors such as hypertension on CVD outcomes may become more complex in patients with a history of early childhood adversities. Such adversities could lead to mental disorders and diabetes, potentially intensifying the interplay between hypertension and CVD outcomes disproportionately. This exemplifies a non-linear dynamic, where the combined effect of these conditions on CVD progression is greater than the sum of their individual effects, particularly in patients with such comorbidity profiles. Building on these observations, it is also important to consider the role of ELFs in these prediction models. While research has identified associations between ELFs and the later development of CVD or diabetes, the extent of their contribution to predictive accuracy is unclear. It is important to distinguish that merely identifying statistical associations between variables, even if they are longitudinal and strong, does not necessarily translate into effective predictors for new observations, underscoring a gap in applying these findings to predictive modeling [16]. In a similar vein, causal inference approaches (e.g., propensity scoring) can offer insights into mechanisms or intervention effects, but they are not designed to optimize predictive performance in complex, comorbid populations. Therefore, the strategic selection and application of these factors are essential for accurate predictions [17]. Machine learning (ML) presents a transformative opportunity to enhance predictive models for CVD and diabetes. ML approaches, capable of processing vast and complex datasets, can identify not only novel risk factors but also intricate, non-linear interactions among them [13]. This capability is expected to improve the accuracy of predictive models [13,18].

Given the dynamics of PCM multimorbidity, we expect that ML surpasses traditional approaches in accurately predicting CVD and diabetes by integrating ELFs and more proximal risk factors as predictors in ML models. In this paper, we leverage ML to investigate the role of ELFs in predicting CVD and diabetes, particularly within comorbid groups. Specifically, we employ XGBoost (XGB) [19] models. The ability of this type of model to handle high-dimensional data makes it appropriate for biomedical studies. This choice is supported by findings from a recent systematic review on type 2 diabetes prediction, where tree-based models were shown to achieve superior predictive performance [20]. Utilizing the UK Biobank (UKB) [21], a comprehensive national health resource from the UK, our objectives are twofold. First, we aim to systematically compare the long-term predictive ability of ELFs for CVD and diabetes among individuals with and without comorbidities. We define these comorbidities as the co-occurrence of mental disorders (depression or anxiety) with either diabetes, in the context of CVD prediction, or with CVD, when predicting diabetes outcomes. Second, we explore the prediction model drivers of these outcomes, including the potential role of ELFs as key contributors, thereby enhancing our understanding of developing multimorbidity. Through this approach, we aim to provide insights that could enhance the precision of existing risk prediction algorithms, specifically tailored for individuals with comorbid conditions. This research incorporates both proximal and early-life risk factors within a large prospective cohort study, rigorously tracking a wide range of variables for each participant. Recent studies have explored the ability of automated ML frameworks to explore wide arrays of non-traditional risk factors. For example, up to 473 and 109 input variables were considered for CVD [13] and diabetes [18] prediction, respectively. However, ELFs were excluded due to missing rates exceeding 50% in patient outcomes. This study conducts a comprehensive

investigation into the value of ELFs in predicting CVD and diabetes within comorbid groups, utilizing statistical and ML techniques. Our study shows that these factors not only enhance the precision of our predictive models but also prove to be key drivers of prediction in these complex patient groups.

## Materials and methods

### Study design and participants

*Selection of dataset.* For this study, we use the UKB for its comprehensive health data, which spans demographics, health status, and disease incidence among a diverse representation of the UK population [21]. The archive includes fully anonymized records from half a million individuals collected from 2006 to 2019. For our analysis, a subset of approximately 150,000 participants without prior diagnoses of CVD or diabetes for which ELFs were available was selected. This allows us to track the onset of these conditions over a decade. S1 Fig illustrates the stratification of these participants, detailing the process for selecting the final study cohorts with CVD and diabetes outcomes.

The UKB data were accessed for research purposes from November 11, 2022, to December 17, 2023. Access to the UKB data for this research is facilitated under the project titled 'Association between Early-Life-Stress and Psycho-Cardio-Metabolic Multi-Morbidity: The EarlyCause H2020 Project' (application number 65769). The UKB obtained ethical approval from the North West Multi-centre Research Ethics Committee (MREC) and the Community Health Index Advisory Group (CHIAG). We adhere to the ethical approval obtained by the UKB through the signature of MTA (Material Transfer Agreement). All participants in this study provided written informed consent.

*Selection of variables.* The study considers a range of ELFs available from our UKB application, including eight ELF variables (both prenatal and postnatal, including exposure to maternal smoking and childhood maltreatment; details in S1 Table), contextualized within a spectrum of demographic and proximal risk factors. Demographic characteristics include age, sex, and ethnicity, while proximal risk factors comprise external exposome data: socioeconomic status (income, qualifications) and physio-metric data (body mass index, height, weight, etc.). They also encompass family history (illnesses of parents or siblings), medical history, lifestyle characteristics (physical activity, diet, sleep habits, smoking, and alcohol use), mental health history (depression, anxiety), and subjective well-being indicators (mood swings, loneliness, etc.). Internal exposome is also available in the form of blood assay results, reflecting the body's internal biological environment. This includes key biomarkers such as Apolipoprotein A, Apolipoprotein B, C-reactive protein, glycated hemoglobin (HbA1c), and random glucose (RG). Note that while HbA1c and glucose are well-established markers for diabetes [22], in our study, they are measured at study entry and treated as baseline predictors. Our models are designed to assess future diabetes risk in an initially non-diabetic population, with individuals diagnosed with diabetes at baseline excluded from the cohort. This approach avoids data leakage and aligns with prior diabetes risk prediction research [18,23, 24]. All variables had a missingness rate of less than 25% among participants with positive diabetes outcomes, corresponding to a missingness rate of 20% for the entire participant population. To address missing data, we applied a statistical imputation algorithm to recover the missing values. S1 Table lists the ELFs along with the other input variables among the 64 considered, grouped by category (e.g., sociodemographics, psychosocial factors, blood assays). These variables were preprocessed into 83 features as a result of one-hot encoding applied to categorical variables with multiple levels.

***Selection of benchmarks.*** The performance of our models is compared with that of well-established risk prediction algorithms: the Framingham score for CVD and DiabetesUK for diabetes. The Framingham score, based on seven core risk factors (sex, age, systolic blood pressure, treatment for hypertension, smoking status, history of diabetes, and BMI), is a widely recognized algorithm for assessing CVD risk [25]. UKDiabetes [26], which incorporates seven features (sex, age, ethnicity, family history, waist size, BMI, and high blood pressure requiring treatment), was selected for assessing diabetes risk due to its alignment with the demographics of the UKB dataset. Framingham and DiabetesUK models are used in our study as benchmarks to visualize the challenges of risk prediction in the comorbid population. Although their performance could be improved through different learning strategies, we decided to use the available models as model optimization is not the primary focus of our study. Additionally, we assess the performance of ML models learned using only the seven core variables used by Framingham score and Diabetes UK for CVD and diabetes risk prediction, respectively. This enables us to separately investigate the influence of employing ML models and the impact of using a broader variable set on prediction accuracy.

***Model and validation strategy.*** In our study, we utilize XGB, an ML algorithm that sets a series of gradient-boosted trees to progressively refine predictions, thereby systematically enhancing model accuracy. Its nonparametric nature contributes to robustness in high-dimensional settings, and the algorithm's ability to recognize complex variable interactions makes it particularly apt for this study. Models are implemented using Python's Scikit-learn library [27], with hyperparameter optimization via the Optuna framework [28]. Note that our goal is *not* solely to develop the best ML frameworks for CVD and diabetes risk prediction possible, but rather to leverage ML models' capacities to reveal latent relationships among variables, particularly to elucidate the association between ELFs and the development of PCM conditions.

To ensure the reported model performance is robust and not unduly influenced by overfitting, we conduct 5-fold stratified cross-validation. Our evaluation metrics include the area under the receiver operating characteristic curve (AUC-ROC) for assessing prediction accuracy, and the Brier score for model calibration. These are separately assessed on non-comorbid and comorbid test groups. It is important to note that a single, unified XGB model is trained to evaluate performance across both the comorbid and non-comorbid subgroups, rather than developing separate models for each subgroup. This approach allows for a consistent evaluation of predictor contributions and comparative performance across the two groups. Additionally, decision curve analysis [29] is performed to assess the clinical utility of models for the test's comorbid subgroup. This analysis calculates the clinical 'net benefit' of prediction models across a range of risk threshold probabilities, $t$, which represents the minimum likelihood at which intervention is considered. Given a threshold $t$, the net benefit $b(t)$ is defined by the difference between the proportion of true positives (TP) and the proportion of false positives (FP) adjusted for $t$:

$$b(t) = \frac{TP}{n} - \frac{FP}{n}\left(\frac{t}{1-t}\right)$$

where $n$ represents the number of samples. For the sake of comparison, we also show the decision curves of two simplistic models: one that predicts risk for all patients, and a model that never predicts risk. Models that yield a higher net benefit at clinically relevant decision thresholds are preferred, as they provide a more advantageous balance between correctly identifying cases and avoiding unnecessary interventions for these groups.

***Outcome.*** In our study, a CVD event is identified through the assignment of ICD-10 diagnosis codes F01 (vascular dementia), I20-I25 (coronary/ischemic heart diseases), I50 (heart

failure events, including acute and chronic systolic heart failures), and I60-I69 (cerebrovascular diseases), or ICD-9 codes 410-414 (ischemic heart disease) and 430-434, 436-438 (cerebrovascular disease), with self-reported data also included. While the diagnosis of vascular dementia can be challenging due to confounding factors, its inclusion follows methodologies from prior large-scale studies to ensure comprehensive coverage of CVD-related conditions [13]. For diabetes, events are identified by the assignment of ICD-10 codes E10 (type 1 diabetes), E11 (type 2 diabetes), and E13/E14 (unspecified diabetes), or ICD-9 code 250 (Diabetes mellitus), along with self-reported data. Gestational diabetes (ICD-10 O24) and diabetes insipidus (ICD-10 E23.2) were not included as they are distinct conditions with separate etiologies and clinical implications. Additionally, comorbid conditions of depression and anxiety are identified based on ICD-10 codes F32 and F33, and ICD-9 code 311 for depression, and ICD-10 code F41 and ICD-9 code 3000 for anxiety, including self-reported instances of these conditions.

## Variable contribution analysis

In this section, we examine the contribution of individual variables to model performance and decision-making, including early-life factors. Specifically, we study their predictive utility for health outcomes and their role in shaping model decisions across subgroups.

*Group-specific Bayes error estimation.* In our analysis, we quantify the performance limits of CVD and diabetes predictive tasks across patient subgroups by approximating the theoretical Bayes optimal classifier, which attains the smallest expected error. This minimal achievable error is also referred to as the Bayes error or irreducible noise, representing the portion of prediction error that cannot be eliminated by any model regardless of the learning algorithm. Estimating group-specific Bayes error provides insights into the inherent difficulty of disease prediction within each subgroup, helping to identify whether certain populations are fundamentally harder to forecast, independent of model choice.

We need to estimate the Bayes error since calculating the real value is unfeasible in general practice. We estimate its upper and lower bounds to narrow it down for each subgroup. The lower bound (Elow) represents the best-case minimal error achievable under ideal conditions, while the upper bound (Eup) reflects a worst-case estimate of the minimal error attainable even by the optimal classifier. The interval drawn with the upper and lower bounds for different subgroups can be compared. Non-overlapping intervals might indicate a different inherent difficulty of prediction for the groups.

We conduct a comparative evaluation on the estimation of Bayes error lower and upper bounds ($E_{low}$ and $E_{up}$) for non-comorbid and comorbid groups, following the methodology outlined in [30], through non-parametric techniques including the Mahalanobis distance [31], Bhattacharyya distance [32], and a k-nearest neighbor method [33] with $k = 5$ and 5-fold cross-validation. This analysis is applied across two distinct sets: a subset of the seven core variables utilized by benchmarks and the whole set of variables, to investigate the benefits of using additional information to make predictions for specific patient subgroups.

*Performance model drivers.* Performance model drivers are variables that significantly impact the model's accuracy, serving as key determinants in achieving optimal predictive performance. These are referred to as *predictive features*. In our study, we focus on evaluating the impact of ELFs and other risk factors on the performance of predictive models for CVD and diabetes. Our goal is to quantify how specific features drive model performance, thereby allowing us to identify robust risk predictors, ultimately leading to enhanced risk prediction performance. To achieve this, we employ a twofold strategy rigorously applied to the test sets to ensure the robustness and generalizability of our findings. We measure (1) the individual

predictive ability of each variable and (2) their synergistic predictive power when combined with others.

*1. Predictive power of individual variables.* To evaluate the individual predictive ability of each variable, we fit XGB models for each of the 64 variables separately. Each model is trained with all the training samples, utilizing a single variable as input. Separate evaluations of these models are carried out for the non-comorbid and comorbid test groups using the AUC-ROC. This comparison allows us to identify which variables have a consistent predictive capacity across different health conditions and which are unique or more influential in one group over the other.

*2. Predictive power of synergistic variables.* While assessing the *predictive power of individual variables* reveals their standalone relevance, we use permutation-based variable importance (PFI) to evaluate their relative influence by measuring the change in the model's performance upon each variable's perturbation. Specifically, we calculate the increase in prediction error resulting from the random shuffling of the variable's values across samples [34]. If a variable is predictive of the outcome, shuffling will weaken its relationship with the outcome and consequently increase the model's error. Variables with no strong relationship with the outcome will see little to no change in error after shuffling. Permutation-based feature importance can underestimate the influence of highly correlated features as the information provided by one of them is still available after shuffling from the other one. To anticipate potential underestimation resulting from highly correlated features, Spearman's rank correlation analysis [35] is conducted on our training data. Our examination confirms that none of the variables exhibit complete correlation, ensuring the robustness of our analysis. We define $PFI_j$ for each variable $j$ as follows:

$$PFI_j = \frac{1}{n}\sum_{i=1}^{n} L(y_i, \hat{f}(x_i)) - L(y_i, \hat{f}(x_i^{sh(j)}))$$

where $n$ represents the number of samples in the group being analyzed, which could be either the non-comorbid (NC) or comorbid (C) test group, $L$ is the loss function used to evaluate the model, $y_i$ is the true outcome for sample $x_i$, and $\hat{f}(x_i)$ and $\hat{f}(x_i^{sh(j)})$ are the model's predictions before and after permuting the $j^{th}$ variable. A large $PFI_j$ value indicates that variable $j$ is a significant contributor to the model's predictive performance.

**Prediction model drivers.** Prediction model drivers are variables that directly influence the model's predictive decisions, thereby determining the specific outcomes of interest. They are also referred to as decision's *explanatory features*. Prediction drivers influence the outcome provided by the model, being it right or wrong, whereas performance drivers boost the right outcome. In our study, we aim to identify and quantify the role of ELFs, alongside other risk factors, in influencing our models' prediction of CVD and diabetes across distinct subpopulations, specifically within the non-comorbid (NC) and comorbid (C) groups. To achieve this, we employ a twofold strategy:

*1. SHAP variable importance.* We integrate SHapley Additive exPlanations (SHAP) values [36], a local model-agnostic interpretation method, to quantify the marginal contribution of each variable $j$ in the prediction of model $\hat{f}$ for instance $x$. In this study, $\hat{f}$ refers to our XGB model. This method helps us understand how individual variables affect specific predictive outcomes, providing insights into their influence on the model's decision-making process at an individual prediction level. The SHAP value is defined as:

$$\phi_j(x) = \sum_{V \subseteq F \setminus \{j\}} \frac{|V|!(|F|-|V|-1)!}{|F|!} [\hat{f}_{V \cup \{j\}}(x_{V \cup \{j\}}) - \hat{f}_V(x_V)]$$

where $F$ is the set of all features and $|F|$ is the number of features. SHAP value is computed for each feature over all possible variable combinations ($V \subseteq F$), effectively capturing the non-linear and interaction effects characteristic of complex biological systems. The local accuracy property of SHAP ensures that the sum of all variable contributions equals the model's prediction, $\hat{f}(x) = E[\hat{f}(x)] + \sum_{j=1}^{|F|} \phi_j(x)$, where $E[\hat{f}(x)]$ is the expected value of the model's output. This decomposition enables the identification of the most influential variables as those with greater contributions to the prediction show higher SHAP value attributions. In this study, global variable importance is evaluated by calculating the average of the absolute SHAP values across instances. We define the SHAP variable importance $I_j$ for each variable $j$ as follows:

$$I_j = \tfrac{1}{n} \sum_{i=1}^{n} |\phi_j(x_i)|$$

where $n$ represents the number of test samples in the group being analyzed, which could be either the non-comorbid (NC) or comorbid (C) group. Through this dual analysis, we identify features integral to both comorbid and non-comorbid states, as well as those exclusive to comorbid conditions. This approach clarifies the significance of ELFs and other variables in their contribution to comorbid profiles.

*2. Dual-group risk effect size.* While *SHAP values* provide a detailed, instance-level understanding of variable contributions, the *risk effect size* offers a complementary perspective on how variables influence risk profiles across sub-groups. Effect size, in statistical terms, refers to the magnitude of the difference between groups in a study (e.g., difference between group means). In our context, risk effect size quantifies the degree to which a variable influences the model's predicted risk distribution between high-risk and low-risk groups for developing CVD and diabetes. We use Cohen's D [37], a statistical metric that quantifies the difference in means between two groups, to measure the variable's distributional shift between high-risk and low-risk groups, as predicted by the model. The larger the Cohen's D value, the greater the distributional shift, and thus, the greater the influence of the feature on the model's risk stratification. We apply this procedure (Algorithm 1) to both non-comorbid and comorbid test groups, separately. Through separate rankings of features based on their Cohen's D values in each group, we discern variables that significantly impact the risk profile, reflecting their distinct contributions across these diverse health contexts.

**Algorithm 1: Dual-group risk effect size analyzer.**

**Input:** S, f, $t = 0.5$   ▷ S: set of samples, $f$: model, $t$: threshold to define low/high-risk patients (default value, 0.5)

**Output:** Variables sorted by descending $\mathcal{D}_j$ value

1: Identify the high-risk and low-risk patients by applying threshold $t$ to $f$'s risk probability predictions.
   $G_{\text{high}} = \{x \in \text{S} \mid f(x) > t\}$,  $G_{\text{low}} = \{x \in \text{S} \mid f(x) \leq 1 - t\}$

2: For each sample subset and feature $j$, compute Cohen's D:
   $\mathcal{D}_j = \left|\bar{X}_{j,\text{high}} - \bar{X}_{j,\text{low}}\right| / s_{j,\text{pooled}}$
   where $\bar{X}_{j,\text{high}} = \frac{1}{|G_{high}|} \sum_{x \in G_{high}} x_j$,  $\bar{X}_{j,\text{low}} = \frac{1}{|G_{low}|} \sum_{x \in G_{low}} x_j$ and

   $s_{j,\text{pooled}} = \left(\left((|G_{\text{high}}| - 1) \cdot s_{j,\text{high}}^2 + (|G_{\text{low}}| - 1) \cdot s_{j,\text{low}}^2\right) / (|G_{\text{high}}| + |G_{\text{low}}| - 2)\right)^{\frac{1}{2}}$
   with $s_{j,\text{high}}^2$ and $s_{j,\text{low}}^2$ being the variances of feature $j$ for $G_{high}$ and $G_{low}$, respectively

3: Sort variables by descending $\mathcal{D}_j$ value

To enhance the robustness of the estimation of the variables' predictive power and their contribution to model decisions, 5-fold stratified cross-validation is used. Rather than relying

on a single train-test split, our results average empirical evidence across different training-test data splits, using the entire dataset for testing. It is important to understand that SHAP values help interpret the influence of each individual variable into the model's predictions (e.g., a variable taking a specific value might be associated in the model with a large predicted probability of disease). However, note that this cannot be interpreted as a measurement of how much each variable boosts the model's predictive performance (e.g., accuracy or AUC). SHAP values have been used in the past to rank the top predictors for diabetes risk in the general population [23], offering insights into key model contributors. We distinguish between prediction and performance drivers and use the appropriate techniques to find them.

***Post-hoc analysis based on dominant performance drivers.*** In addition to the primary analyses, we conducted exploratory, post-hoc evaluations to further characterize the contribution of variables that emerged as dominant performance drivers. In particular, HbA1c is identified as a key performance driver for diabetes prediction, especially within the comorbid population. HbA1c is a well-established clinical marker for diabetes diagnosis. In our study, it is measured at patient entry, prior to disease onset, enabling us to assess its predictive value as a baseline risk factor.

To understand how HbA1c contributes to performance in this group and to assess its interaction with other predictors, we perform a post-hoc stratified analysis based on baseline HbA1c values. The cohort of 149,847 participants was divided into two subgroups: a low-normal subgroup (HbA1c < 5.34%) and a high-normal subgroup (HbA1c ≥ 5.34%), where 5.34% is the mean HbA1c value in the whole population [22]. In this analysis, we evaluate model performance across different predictor sets. These include: (1) DiabetesUK, (2) XGB (7 core variables), (3) XGB (HbA1c only), (4) XGB (all variables except HbA1c), and (5) XGB (all variables). Each model is evaluated using 5-fold cross-validation, and AUC-ROC metrics are calculated for the HbA1c-defined subgroups.

To summarize the analytical approach and goals addressed in this study, we provide a workflow diagram (Fig 1). This figure outlines the main analytical components: subgroup

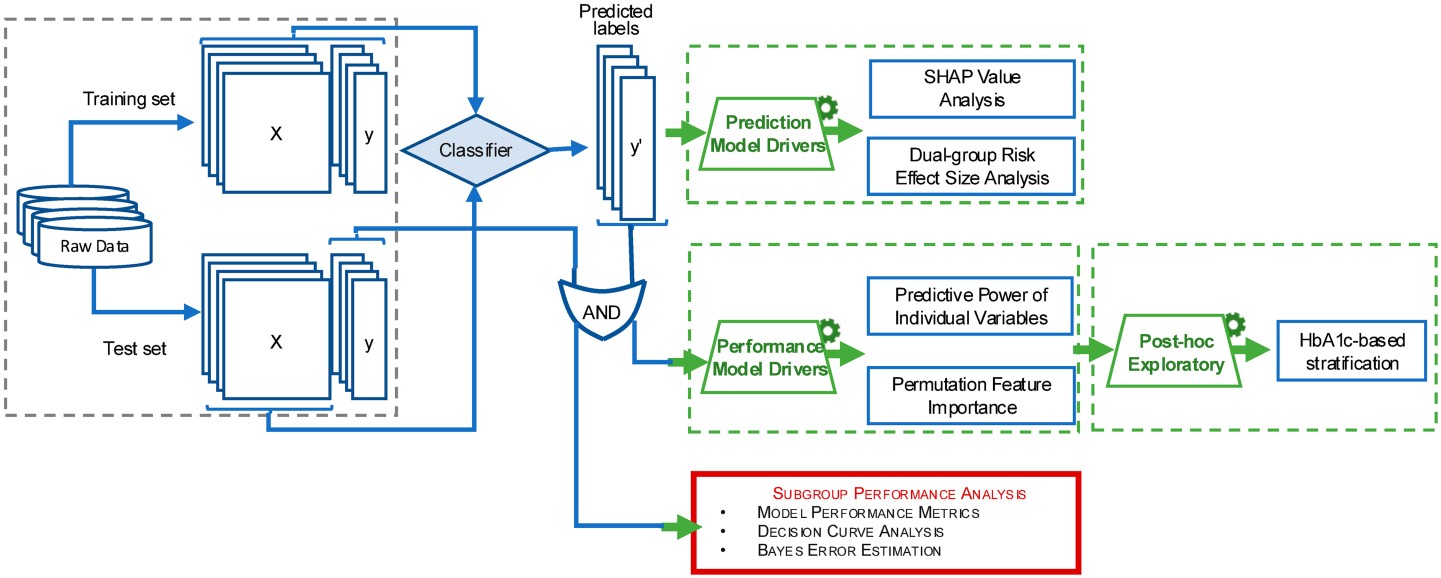

**Fig 1. Overview of the analytical framework characterizing the contribution of early-life and proximal factors in ML-based risk prediction of psycho-cardio-metabolic multimorbidity.**

performance analysis, characterization of performance and prediction model drivers, and post-hoc exploratory analyses. All implementation details, including code and library versions, are available at https://github.com/ngoc-vien-dang/ELF.

## Results

### Predictive performance assessment

**Comparative performance of prediction models.** The predictive performance of the different models, validated separately on non-comorbid and comorbid populations, is illustrated in Table 1. The baseline models, the Framingham score, DiabetesUK, and XGB (7 core variables), show reduced accuracy in the comorbid population, while the XGB (all variables) models exhibit enhanced performance for this population. In the non-comorbid group, the XGB (all variables) models also outperform the baselines in terms of AUC-ROC. The Brier score, shown in Table 1 too, aligns with these results, indicating a consistently better performance of the XGB (all variables) models across both populations.

Note that the XGB (7 core variables) models still achieve an improvement over the Framingham score and DiabetesUK indicating that model selection can boost the performance of a model based on the 7 core features. For CVD prediction, the XGB (7 core variables) model improves the AUC-ROC by +1.73% and +4.83% over the Framingham score in the non-comorbid and comorbid groups, respectively. Similarly, for diabetes prediction, it improves the AUC-ROC by +0.92% and +3.07% over DiabetesUK in the non-comorbid and comorbid groups, respectively.

**Comparative decision curve analysis of prediction models.** Further augmenting our understanding, decision curve analysis in Fig 2 compares the net benefit of our XGB models with that of the benchmark models (an XGB model with the 7 core variables, Framingham model for CVD, and DiabetesUK model for diabetes) in the comorbid population. The plots also include the behavior of two simplistic predictions: a dummy model that always predicts that all patients will develop the condition ('all'), and another one that always predicts that none patient will ('none'). The performance of benchmark models, closely aligned with the 'all' strategy, suggests that they may overestimate risk, potentially leading to a high false positive rate in the comorbid population. The analysis shows that XGB (all variables) models provide greater net benefit than all three benchmark models across a broad range of risk thresholds for both CVD and diabetes prediction.

Specifically in CVD, the XGB (all variables) model begins to deliver a greater net benefit than both the 'all' strategy and the Framingham score when the decision threshold exceeds 0.2, while the XGB (7 core variables) model follows closely. For diabetes, the XGB (all variables) model outperforms both the 'all' strategy and DiabetesUK at a decision threshold above 0.1, and the XGB (7 core variables) model shows a similar trend. Additionally, the XGB (all variables) models for both CVD and diabetes prediction surpass the net benefit of the 'none' strategy across all thresholds, confirming their effectiveness in predicting outcomes for comorbid groups. Next, we analyze the group-specific Bayes error estimates to contextualize the results from Table 1.

### Group-specific Bayes errors

Group-specific Bayes error estimates presented in Table 2 for both CVD and diabetes prediction tasks across comorbid and non-comorbid populations indicate consistently higher

**Table 1. Comparison of models' predictive performance for CVD and Diabetes with and without comorbid conditions.**

| Outcome | Model | Non-Comorbid | | | | Comorbid | | | |
|---------|-------|--------------|---|---|---|----------|---|---|---|
| | | AUC-ROC | | Brier score | | AUC-ROC | | Brier score | |
| | | Value (↑) | Change | Value (↓) | Change | Value (↑) | Change | Value (↓) | Change |
| CVD | Framingham score | 0.70 ± 0.00 | Baseline | 0.08 ± 0.00 | Baseline | 0.55 ± 0.03 | Baseline | 0.22 ± 0.01 | Baseline |
| | XGB (7 core variables) | 0.72 ± 0.00 | +1.73% | 0.07 ± 0.00 | -1.48% | 0.60 ± 0.02 | +4.83% | 0.14 ± 0.01 | -7.36% |
| | XGB (all variables) | 0.73 ± 0.00 | +2.8% | 0.06 ± 0.00 | -1.86% | 0.63 ± 0.02 | +7.9% | 0.14 ± 0.01 | -7.34% |
| Diabetes | DiabetesUK | 0.78 ± 0.01 | Baseline | 0.09 ± 0.00 | Baseline | 0.70 ± 0.01 | Baseline | 0.18 ± 0.01 | Baseline |
| | XGB (7 core variables) | 0.79 ± 0.00 | +0.92% | 0.03 ± 0.01 | -6.39% | 0.73 ± 0.01 | +3.07% | 0.10 ± 0.01 | -8.01% |
| | XGB (all variables) | 0.87 ± 0.00 | +9.07% | 0.03 ± 0.00 | -6.57% | 0.82 ± 0.03 | +12.3% | 0.09 ± 0.02 | -8.68% |

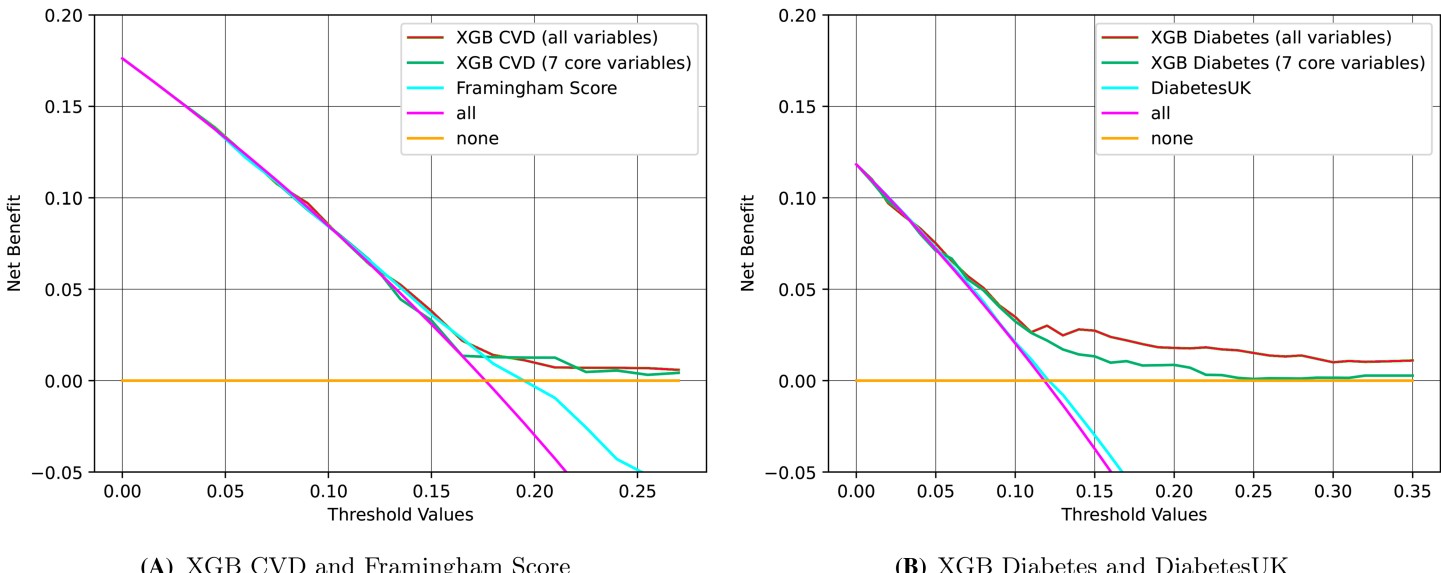

**(A)** XGB CVD and Framingham Score  **(B)** XGB Diabetes and DiabetesUK

**Fig 2. Decision curve analysis for comorbid groups.** Net benefit is provided for different models in (A) CVD prediction and (B) diabetes prediction. The different models include the corresponding benchmarks (blue) –Framingham Score for CVD in (A) and DiabetesUK in (B)–, an XGB model with all variables (green), an XGB model with the corresponding 7 core variables (cyan), and baseline 'all' (purple) and 'none' (orange) strategies

noise estimates for comorbid groups than non-comorbid groups, utilizing two distinct subsets: 7 core variables and all variables. Particularly, nearest neighbor estimates provide non-overlapping intervals between non-comorbid and comorbid groups, and the one for comorbid subgroup is consistently higher. This suggests that the prediction task is more difficult among individuals from this population. Additionally, noise estimates for CVD are consistently higher relative to those for diabetes. These findings align with the performance differences of models shown in Table 1, which shows higher predictive performance for diabetes tasks than for CVD across both subgroups. The Bayes error estimates with the three techniques show similar trends. Estimations based on Bhatttacharyya's method considering all the variables are not provided due to singular matrix errors during their computation.

The demonstrated efficacy of the XGB (all variables) models lays the groundwork for employing post-hoc approaches, previously outlined, for variable ranking.

**Table 2. Bayes error's lower and upper bounds estimation ($E_{low}$ and $E_{up}$) of comorbid and non-comorbid groups for CVD and diabetes prediction. Mahalanobis distance estimates provide only upper bounds; Some Bhattacharyya's estimates are not available due to singular matrix errors.**

| Outcome | Method | Var. set | Non-Comorbid | | Comorbid | |
|---|---|---|---|---|---|---|
| | | | $E_{low}$ ($\downarrow$) | $E_{up}$ ($\downarrow$) | $E_{low}$ ($\downarrow$) | $E_{up}$ ($\downarrow$) |
| CVD | Mahalanobis | 7 core vars. | – | 0.122571 | – | 0.282146 |
| | | All vars. | – | 0.121617 | – | 0.238344 |
| | Bhattacharyya | 7 core vars. | 0.053802 | 0.225627 | 0.152879 | 0.359871 |
| | | All vars. | 0.025346 | 0.157174 | – | – |
| | Nearest Neighbors | 7 core vars. | 0.038402 | 0.073854 | 0.110000 | 0.194059 |
| | | All vars. | 0.037158 | 0.071600 | 0.105107 | 0.188119 |
| Diabetes | Mahalanobis | 7 core vars. | – | 0.047010 | – | 0.194003 |
| | | All vars. | – | 0.043798 | – | 0.154115 |
| | Bhattacharyya | 7 core vars. | 0.015062 | 0.121801 | 0.076367 | 0.265584 |
| | | All vars. | 0.001459 | 0.038166 | – | – |
| | Nearest Neighbors | 7 core vars. | 0.013230 | 0.026110 | 0.070000 | 0.127923 |
| | | All vars. | 0.012713 | 0.025102 | 0.063133 | 0.118294 |

## Key drivers of model performance

**Predictive power of individual variables in comorbid and non-comorbid groups.**
The analysis of the *predictive power of individual variables* is summarized in a scatter plot (Fig 3) comparing the individual predictive ability of the variables for comorbid versus non-comorbid groups, separately for CVD and diabetes. To improve visual interpretability and enable consistent comparison, the horizontal and vertical ranges are delimited to a lower bound based on the minimum value observed among ELF variables and an upper bound at the 99th percentile of the distribution of values from both conditions. Each point represents a variable and it is located according to the performance of a model learned exclusively with this predictive variable on the C and NC groups (x and y axis, resp.). The diagonal line represents single-variable models that show the same performance on both subgroups. For both conditions, variables tend to have a higher predictive ability in the non-comorbid group, as most points concentrate above the line of similar effect. In CVD prediction, *blood assays* and *sociodemographic* factors exhibit the highest predictive ability for both non-comorbid and comorbid groups but are more predictive for non-comorbid groups. In diabetes prediction, *blood assays*, along with *physical measures and activities*, display a similar pattern, demonstrating stronger predictive ability in non-comorbid groups. The complete distribution of AUC-ROC values for all the features is available in S4 Fig.

The distribution of ELFs varies, with some positioned above or near the line of similar effect, reflecting a mixed predictive influence on both comorbid and non-comorbid groups. However, a significant number of ELFs, notably *breastfed as a baby* (AUC-ROC: 0.54 for comorbid, 0.51 for non-comorbid) for CVD and *felt loved as a child* (AUC-ROC: 0.55 for comorbid, 0.52 for non-comorbid) for diabetes are located below the line, suggesting their particular relevance and contribution to the predictive models for comorbid conditions. These ELFs, along with other variables positioned below the line, could explain the enhanced performance of the XGB models for these groups as detailed in Table 1. Note that the AUC-ROC values of these ELFs slightly exceed 0.5, suggesting that their individual utility in the long-term prediction of CVD and diabetes, while present, is limited. However, their higher relevance in the comorbid group suggests they provide added value for risk stratification within this population. Additionally, any factor with modest individual AUC-ROC values can still be

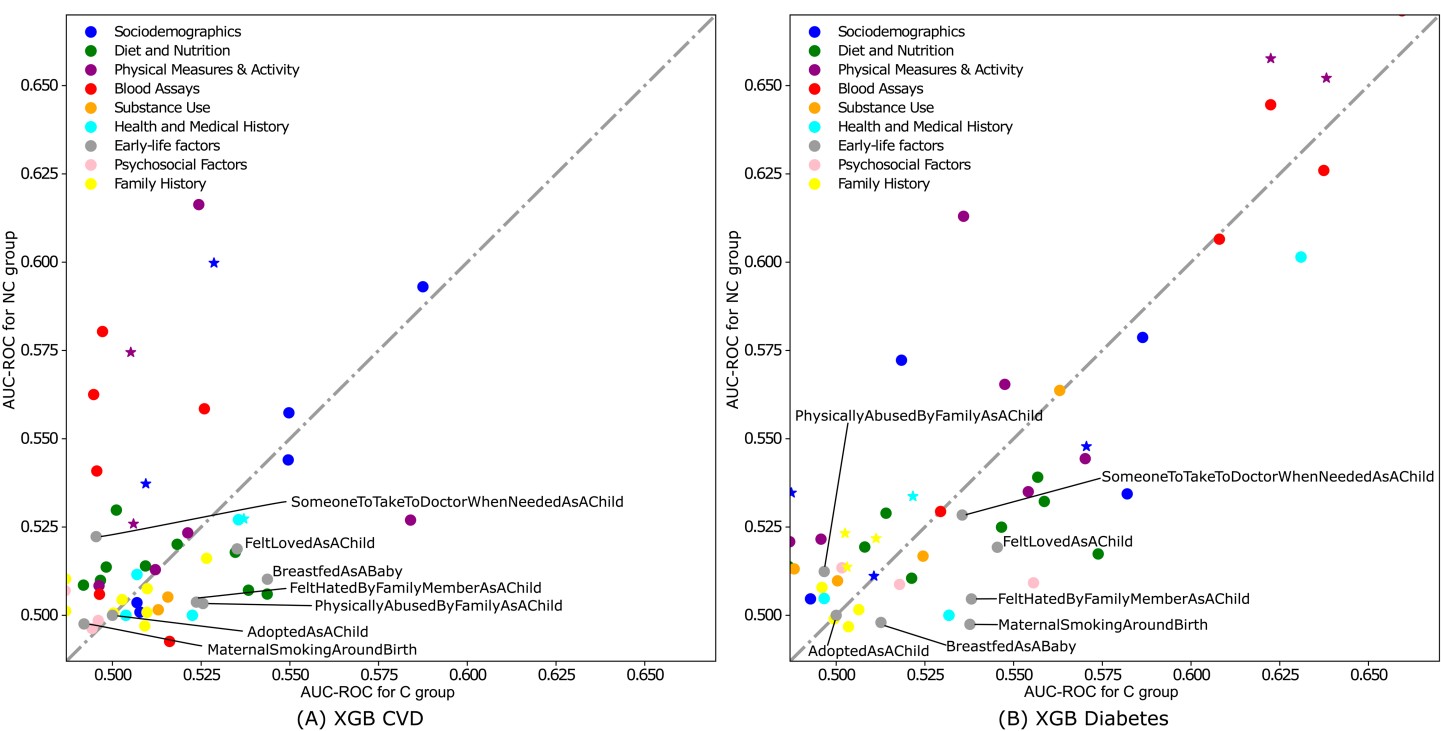

**Fig 3. Predictive ability of individual variables for (A) CVD and (B) diabetes prediction.** Each plot compares AUC-ROC performance in the comorbid group (x-axis) vs performance in the non-comorbid (y-axis) group. Each point represents an XGB model learned with a single feature. Colors differentiate models learned with a variable of different type. The star marker indicates models learned with a variable used by the benchmarks. The dotted grey line indicates equal performance in both groups. Deviation from this line allows for identifying variables that have a stronger predictive impact on one group over the other.

useful for ML models when combined with other features, especially in complex populations like comorbid groups.

**PFI analysis of predictive variables in comorbid and non-comorbid groups.** The *PFI* analysis underscores the predictive ability of age for CVD and HbA1c for diabetes. Table 3 shows results for the comorbid group, whereas details on the non-comorbid group can be found in S2 Table and S3 Table. These findings align with the results of previous works [13, 18]. Our analysis reveals that while there is a notable overlap in the most relevant predictors of CVD and diabetes in both non-comorbid and comorbid groups, with 60% of the top 30 variables being common to both populations, there are specific variables that distinctly influence the comorbid group. Traditional risk factors, such as *smoking status* in the case of CVD and *waist size* for diabetes, maintain their predictive validity in the non-comorbid group; however, their predictive power diminishes in the comorbid group for both conditions. Furthermore, *self-reported overall health rating*, which is not usually considered in existing risk prediction models, and clinical measures such as *metabolic factors*, show more predictive power for individuals with comorbid conditions. This variation aligns with the findings in Table 1, where the XGB models learned with all the variables demonstrate superior performance over benchmark models, especially for the comorbid group.

As shown in S2 Table and S3 Table, the analysis indicates an elevated prevalence and ranking of ELFs in the comorbid group for both CVD and diabetes, indicating their interdependent contribution to the model's overall predictive performance in patients with multiple health issues. Specifically, in CVD prediction, 4 ELFs rank in the top 30 for comorbid groups

**Table 3. Permutation-based variable ranking for predicting CVD and diabetes in comorbid groups.**

| Variable (CVD prediction) | Score | Variable (Diabetes prediction) | Score |
|---|---|---|---|
| Age*† | 0.056253 | HbA1c† | 0.119222 |
| OverallHealthRating† | 0.016420 | WaistCircumference*† | 0.023598 |
| Glucose† | 0.007265 | OverallHealthRating† | 0.010247 |
| CurrentEmploymentStatus_Retired | 0.007114 | Triglycerides† | 0.006574 |
| WaistCircumference† | 0.006805 | HDLCholesterol† | 0.005897 |
| HbA1c† | 0.006483 | Glucose† | 0.003622 |
| CurrentEmploymentStatus_Paid/SelfEmployed† | 0.003173 | Hypertension*† | 0.003426 |
| LDLDirect† | 0.003124 | Sex† | 0.003144 |
| IllnessesOfMother_CVD† | 0.002986 | BodyFatPercentage† | 0.002629 |
| SmokingStatus*† | 0.002780 | Age*† | 0.001551 |
| DiastolicBloodPressure† | 0.002697 | CurrentEmploymentStatus_Sick/Disabled | 0.001472 |
| Depression | 0.002218 | Qualifications_UnivDegree† | 0.001422 |
| CurrentEmploymentStatus_Sick/Disabled† | 0.002155 | BMI*† | 0.001389 |
| PorkIntake | 0.001957 | SomeoneToTakeToDoctorWhenNeededAsAChild | 0.000909 |
| CReactiveProtein† | 0.002057 | CerealIntake | 0.000909 |
| PhysicallyAbusedByFamilyAsAChild | 0.002031 | FeltHatedByFamilyMemberAsAChild | 0.000865 |
| Cholesterol† | 0.001815 | SleepDuration | 0.000858 |
| ApolipoproteinA† | 0.001792 | AvgHouseholdIncome† | 0.000729 |
| BodyFatPercentage† | 0.001593 | RawVegetableIntake | 0.000705 |
| MaternalSmokingAroundBirth | 0.001591 | ProcessedMeatIntake† | 0.000691 |
| SaltAddedToFood | 0.001431 | Depression | 0.000648 |
| SleepDuration | 0.001137 | Qualifications_NoneAbove | 0.000625 |
| AlcoholIntakeFrequency | 0.000868 | IllnessesOfMother_Diabetes*† | 0.000591 |
| WaterIntake | 0.000839 | IllnessesOfSiblings_Diabetes*† | 0.000532 |
| FeltHatedByFamilyMemberAsAChild† | 0.000759 | CReactiveProtein† | 0.000442 |
| BreastfedAsABaby | 0.000708 | IllnessesOfFather_CVD | 0.000434 |
| Qualifications_UnivDegree† | 0.000623 | FeltLovedAsAChild | 0.000416 |
| HDLCholesterol† | 0.000580 | AlcoholIntakeFrequency† | 0.000394 |
| MoodSwings | 0.000530 | PoultryIntake | 0.000387 |
| Qualifications_ProfQual(Nurse/Teach) | 0.000517 | IllnessesOfMother_CVD | 0.000347 |

*Variables used by the benchmarks. † Shared variables between comorbid and non-comorbid groups. Underlined variables indicate ELFs.

among 83 features, versus 2 in non-comorbid groups (with lower placement), and one overlapping ELF between groups. For diabetes, 3 ELFs make the top 30 in comorbid groups, against a single, lower-ranked ELF in non-comorbid groups. This finding is consistent with the detailed predictive ability of individual variables analysis presented in Fig 3, confirming the significant role of ELFs in comorbid scenarios.

## Key drivers of model prediction

In this section, we analyze the variables central to predicting the risk of CVD and diabetes, focusing on the contrasting profiles between non-comorbid and comorbid groups. This analysis provides detailed explanations of what led to a patient's predicted risk, as determined by ML-based models.

**SHAP value analysis of predictive variables in comorbid and non-comorbid groups.** *CVD prediction.* For those with comorbid conditions, as shown in Fig 4, *age*, *sex*, and the presence of *diabetes* are the most significant variables for predicting the risk of CVD, indicating a higher risk profile in these patients. Additionally, metabolic health measures such as *waist circumference* and *LDL cholesterol* play significant roles, underscoring that the learning

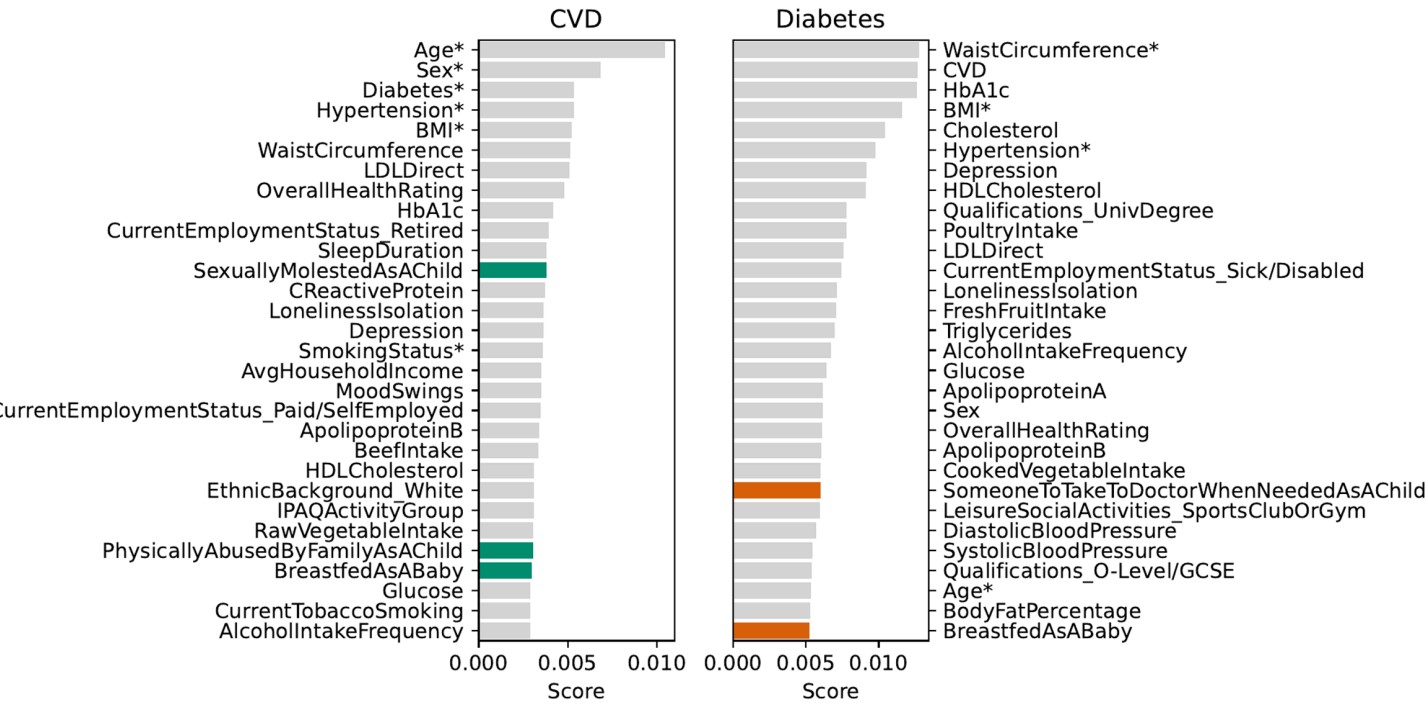

**Fig 4. Averaged SHAP scores for the top 30 influential variables in the prediction models for CVD and diabetes within comorbid patient groups.** Core variables used by the benchmark models are marked with an asterisk (*). Coloured bars highlight ELFs.

method has assigned them a large influence in the model predictions, which is more notorious in the case of comorbidities. Notably, psychosocial experiences, captured by variables reflecting *childhood abuse*, *feelings of isolation*, and *depressive symptoms*, are identified as powerful predictors. This would support the integration of psychological health considerations into a comprehensive CVD risk assessment, especially for patients managing coexisting conditions. In the non-comorbid cohort, our *SHAP value* analysis for CVD prediction, detailed in S5 Fig, underscores the impact of personal lifestyle factors, including *alcohol intake frequency* and *smoking status*, as well as *social engagement*, *overall health ratings*, and *body fat percentage*, as significant drivers of risk prediction. Traditional risk factors such as *age* and *blood pressure* retain their expected prominence. These results emphasize the importance of lifestyle choices in developing CVD risk for individuals without additional health conditions.

*Diabetes prediction.* In the context of diabetes prediction, the analysis presented in Fig 4 for the comorbid group underscores an amplified significance of metabolic factors, with *waist circumference* and *HbA1c* demonstrating higher SHAP scores. These findings align with established clinical understanding, demonstrating that the model has accurately captured key risk factors for comorbid patients in a clinically relevant manner. This trend points to the significant role of metabolic dysregulation in diabetes risk in individuals with multiple health challenges. Additionally, the presence of CVD is identified as a key predictor. The SHAP scores also highlight the critical role of psychosocial factors, particularly *depression*, indicating the evident interplay between mental and physical health in diabetes management for comorbid patients. Furthermore, ELFs like *having someone to take to the doctor when needed as a child* and *being breastfed as a baby* are found to have enduring impacts on health outcomes. These

findings advocate for a diabetes risk model that integrates a diverse range of physiological, psychological, and ELFs, particularly pertinent in the context of comorbidity.

For individuals without comorbidities, *glucose levels* unsurprisingly dominate as a key factor, followed by traditional clinical markers such as *HbA1c* and *waist circumference*, reflecting their critical roles in diabetes risk prediction. Beyond these, lifestyle habits including *smoking status* and dietary patterns, particularly the consumption of *cooked vegetables* and *fresh fruit*, are identified as influential, which underscores the importance of day-to-day health behaviors in diabetes risk prediction in the comorbid population. Note that the referred dietary factors included in this study reflect adult dietary behaviors, rather than early-life dietary exposures.

Poultry intake also ranked as a prominent risk predictor for diabetes, placing 10th in our SHAP analysis. This result aligns with recent evidence from a multi-regional study, which observed a positive association between poultry intake and diabetes risk, particularly in European populations, including the UK [38]. However, the association was less consistent in regions such as South Asia and the Eastern Mediterranean. Given poultry's classification as a healthier alternative to red meat, these results suggest the need for further investigation into its role in diabetes risk prediction. Notably, ELFs such as feeling loved during childhood, although less dominant, still appear in the analysis, suggesting that the learning technique is uncovering a subtle yet present impact on long-term health outcomes. However, it is crucial to acknowledge that these findings, particularly from self-reported ELFs like childhood maltreatment, may be influenced by reporting biases, especially in individuals with mental health issues who may be more likely to negatively appraise their life histories, potentially influencing associations between these factors and health outcomes. For a detailed analysis of the non-comorbid cohort, see S6 Fig.

**Dual-group risk effect size analysis of predictive variables.** In the assessment of explanatory variables using the *dual-group risk effect size analyzer*, Fig 5 shows a substantial overlap of approximately 60% in the top-30 variables between non-comorbid and comorbid groups for both CVD and diabetes. *Age* and *HbA1c* are consistently the top-ranking variables for these groups in CVD and diabetes, respectively (see S4 Table and S5 Table for a list of variables ordered by their effect size). In the comorbid group, ELFs as well as psychosocial factors such as *mood swings* and *irritability*, which are generally underrepresented in traditional risk models, display a significant effect size for both CVD and diabetes prediction. This may be explained by the possibility that these indicators reflect underlying depression severity, influencing the likelihood of developing comorbid conditions like CVD and diabetes. This indicates that our models rely on emotional well-being and formative childhood experiences for disease risk prediction among these individuals. The *dual-group risk effect size* analysis thus complements the *SHAP value* analysis by providing additional insights into how these variables, especially ELFs, distinctly influence risk stratification in comorbid scenarios.

Expanding upon these findings, we observe that the association between ELFs and PCM multimorbidity can be modified by lifestyle factors. This association seems to be positively moderated by health-promoting behaviors such as increased intake of *fresh fruits*, and both *raw and cooked vegetables*, alongside engagement in *leisure social activities* and consistent *physical activity*. Conversely, the association is exacerbated by negative lifestyle habits, including *smoking* and *alcohol consumption*, along with psychological factors like *mood swings* and *feelings of loneliness*. A causal analysis, which is beyond the scope of this work, would be required to find causal relationships between these behaviors and the respective risk outcomes.

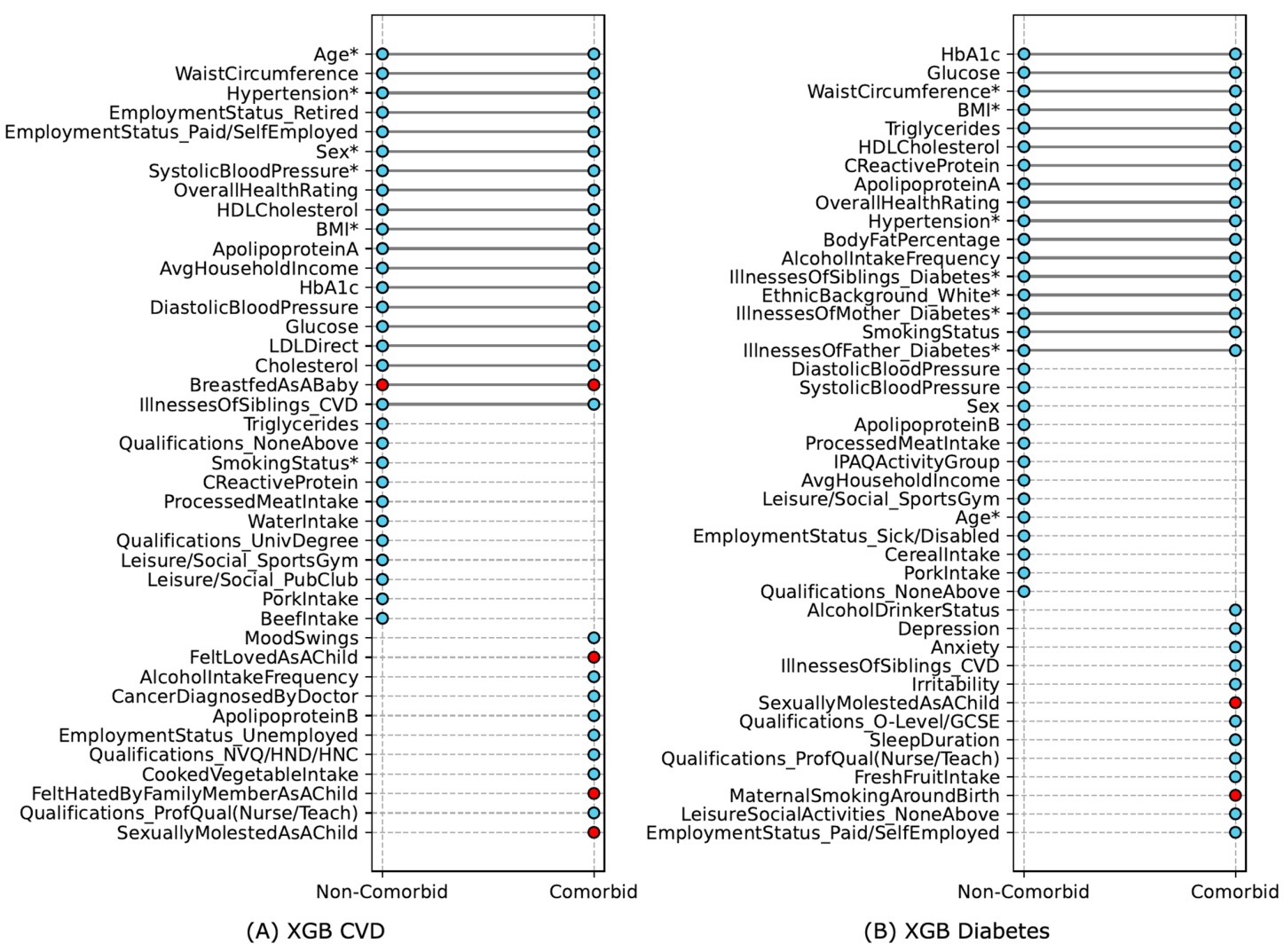

**Fig 5. Differential top features by effect size: comorbid vs. non-comorbid.** The top 30 features are identified by their effect size for (A) CVD and (B) Diabetes predictive models. All features are indicated by blue dots, and red dots specifically highlight ELFs. Variables marked with an asterisk (*) are used in benchmarks.

## HbA1c's role in diabetes prediction for the comorbid cohort

As outlined in Section Materials and methods, a post-hoc stratified analysis was performed to evaluate the role of HbA1c in diabetes prediction within the comorbid population, following its identification as a dominant performance driver (Table 3). As shown in Table 4, the predictive performance of XGB models for the comorbid cohort varies across HbA1c-defined subgroups. In the low-normal HbA1c group, XGB models trained on 7 core variables or non-laboratory variables achieve high predictive accuracy, demonstrating the model's ability to predict diabetes risk without relying on HbA1c. By contrast, the XGB model trained only on HbA1c shows limited predictive value for this group. In the high-normal HbA1c group, HbA1c plays a stronger role, with its inclusion improving the model's performance compared to using all predictors except HbA1c. The XGB model trained only with HbA1c also outperforms the benchmarks (XGB trained on 7 core variables and DiabetesUK). However, HbA1c is not the sole performance driver in this group. The XGB model trained only on

**Table 4. Comparison of models' predictive performance (AUC-ROC) for diabetes risk in HbA1c-defined subgroups within the comorbid group.**

| Method | HbA1c < 5.34% | HbA1c ≥ 5.34% |
|---|---|---|
| DiabetesUK | 0.83 ± 0.05 | 0.66 ± 0.02 |
| XGB (7 core variables) | 0.88 ± 0.06 | 0.70 ± 0.02 |
| XGB (HbA1c) | 0.51 ± 0.10 | 0.75 ± 0.04 |
| XGB (all variables - HbA1c) | 0.87 ± 0.04 | 0.74 ± 0.02 |
| XGB (all variables) | 0.87 ± 0.05 | 0.81 ± 0.04 |

HbA1c achieves lower predictive performance than the model trained on all predictors. Additionally, the model trained on all predictors except HbA1c demonstrates competitive performance. These results underscore HbA1c's critical role as a performance driver, particularly in the high-normal group, while highlighting the importance of integrating diverse predictors to achieve robust performance across the whole population of comorbid patients.

Fig 6 displays the feature importances as performance drivers in the XGB model with all the features across HbA1c-defined subgroups. In the high-normal group, HbA1c ranks as the top feature by FPI score, while in the low-normal group, it ranks second, with waist circumference as the leading contributor. The relative differences in variable scores suggest that the model relies on many predictors other than HbA1c to maintain performance across both subgroups.

The role of HbA1c as a prediction driver is slightly different, as shown in Fig 7. In the high-normal HbA1c group, HbA1c ranks as the top feature, with the largest effect size among all predictors, reflecting its prominent role in shaping individual-level predictions. In the low-normal HbA1c group, HbA1c ranks sixth, with predictors such as waist circumference, BMI, and overall health rating ranking higher. Notably, laboratory-based features are key prediction drivers in the high-normal group, while ELFs consistently feature among the top predictors in both subgroups, reflecting their long-term impact on diabetes risk. These results emphasize the importance of using clinical laboratory markers and early-life information to support risk stratification and preventive strategies in the comorbid group.

## ELF contribution for diabetes prediction with smaller predictor sets

Diabetes prediction can achieve high accuracy with a minimal number of predictors, as few as four biomarkers, illustrating a balance between prediction accuracy and the cost of acquiring additional variables in clinical practice [39–41]. In this line, we designed an experiment aimed at identifying a compact set of predictors that preserves high predictive performance and evaluates ELF contributions within this set. We start with the seven core UKDiabetes' features plus all the ELFs. Incrementally, we add other features to the model based on their PFI scores in the high-normal HbA1c group. This strategy allowed us to assess how the inclusion of each variable influenced the model's predictive performance.

We observe that the AUC-ROC improved rapidly with the first 20 variables, which include the 7 core variables and ELFs, but plateaued after the inclusion of approximately 25 predictors. Beyond this point, additional features provided only marginal gains. This result suggests that a focused subset of around 25 predictors achieves an optimal balance between predictive utility and simplicity, leading to further evaluation of ELF contributions in this model. This 25-feature model achieves an AUC-ROC of 0.90 ± 0.06 (95% CI) for the non-comorbid group and 0.82 ± 0.03 (95% CI) for the comorbid group. These results are slightly better than those achieved by the XGB (all variables) model in Table 1, demonstrating its ability to balance predictive accuracy and simplicity effectively.

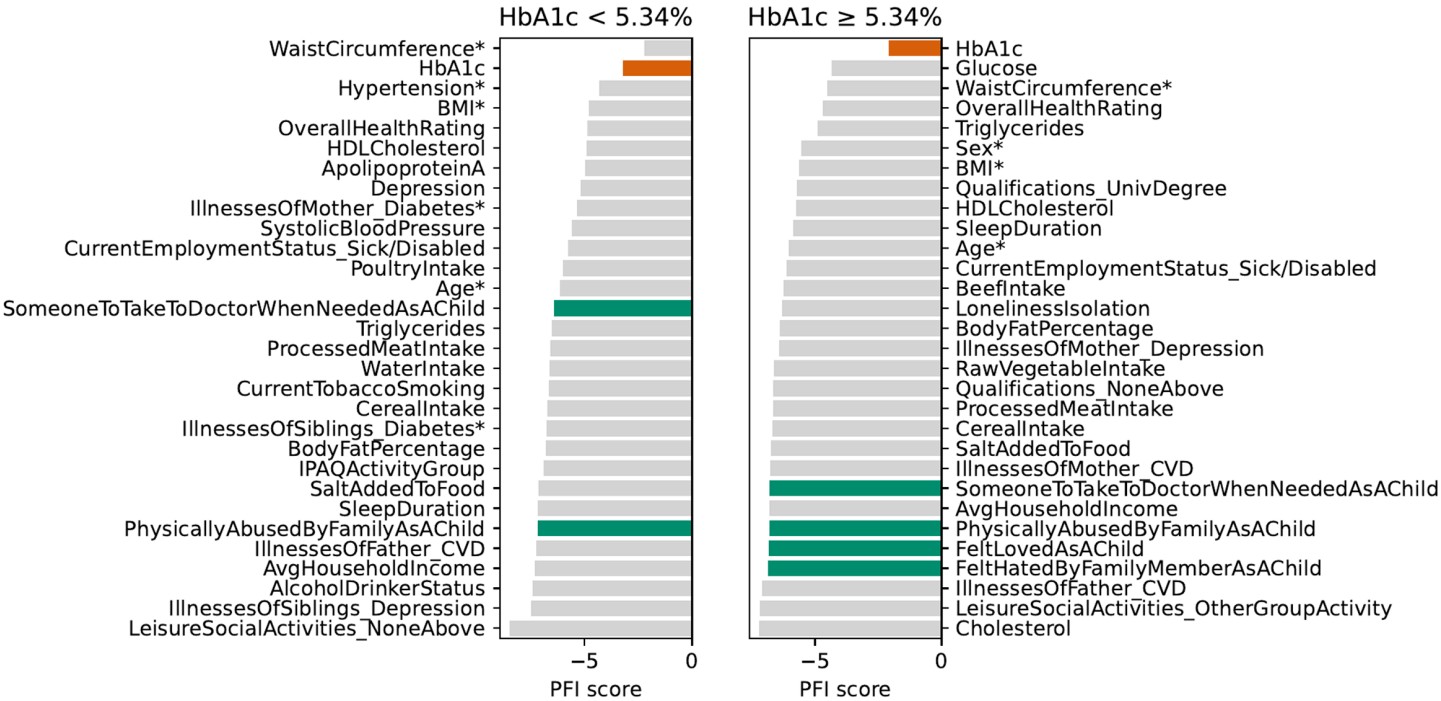

**Fig 6. Top 30 features for XGB (all variables) by FPI score in HbA1c-defined cohorts within the comorbid group.** Core variables used by the benchmark models are marked with an asterisk (*). Green bars highlight ELFs; orange bars indicate HbA1c. Note that PFI scores are log-transformed to enhance the visibility of smaller values while maintaining relative differences.

We also examine ELF contributions in the absence of lab-based predictors. This analysis aims to assess the relative importance of ELFs when costly laboratory features are excluded, relying only on easy-to-collect variables. Thus, we learned an XGB model only with non-laboratory variables and it achieved an AUC-ROC of 0.89 ± 0.07 (95% CI) for the non-comorbid group and 0.73 ± 0.03 (95% CI) for the comorbid group.

ELF contributions consistently increase in both the top 25-variable model and the non-laboratory variable model relative to the all-variable model, as summarized in Table 5. Notably, SHAP analysis shows that ELF contributions rise from 6.97% in the all-variable model to 16.70% in the top 25-variable model, representing an absolute gain of 9.74 percentage points (a 139.77% relative increase). The non-laboratory variable model also exhibits higher ELF contributions, reinforcing their role as key predictors in the comorbid population when laboratory biomarkers are not available.

## Discussion

### Key findings

In a comprehensive analysis based on the UKB, our first key finding is that the ML model, trained with data from 150,000 participants and 64 variables, demonstrates improvement in predicting the 10-year risk of CVD and diabetes compared to traditional risk assessment tools, specifically the Framingham score and the DiabetesUK model, respectively. Our analysis shows that traditional risk models, while demonstrating utility in the general population,

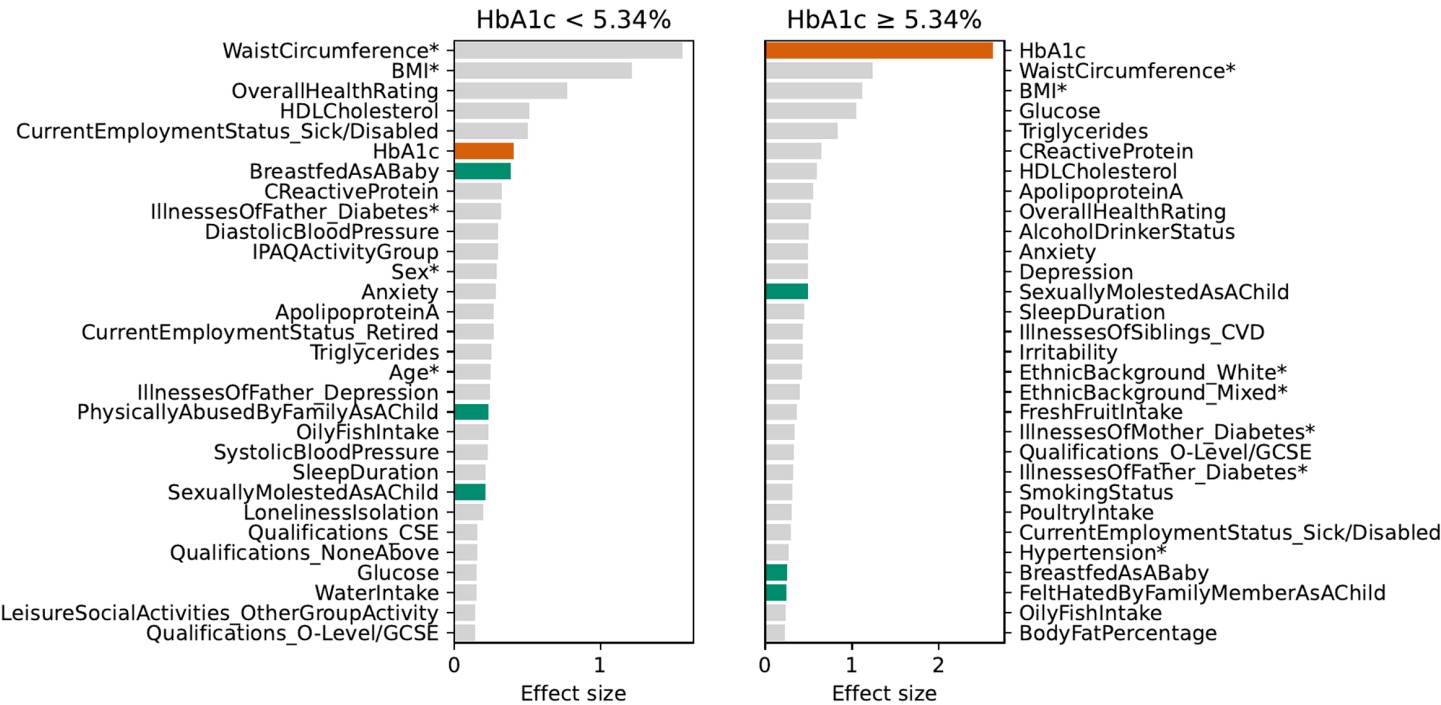

**Fig 7. Top 30 features for XGB (all variables) by risk effect size in HbA1c-defined cohorts within the comorbid group.** Core variables used by the benchmark models are marked with an asterisk (*). Green bars highlight ELFs; orange bars indicate HbA1c.

**Table 5. ELF accumulated contributions, measured by PFI, SHAP, and effect size, for the comorbid group in XGB models learned with all variables, non-laboratory variables, and the top 25 variables. Normalized values ensure comparability across models. Gain refers to the absolute and relative increases in ELF contributions compared to the all-variable baseline model. Absolute gain is expressed as the percentage point difference, while relative gain (in parentheses) is calculated as the percentage increase relative to the baseline value.**

| Model | PFI | | SHAP | | Effect size | |
|---|---|---|---|---|---|---|
| | Value (%) | Gain | Value (%) | Gain | Value (%) | Gain |
| XGB (all vars) | 0.45 | Baseline | 6.97 | Baseline | 5.11 | Baseline |
| XGB (Top 25 variables) | 2.91 | +2.91 (547.58) | 16.70 | +9.74 (139.77) | 9.13 | +9.74 (139.77) |
| XGB (non-lab. variables) | 2.24 | +1.79 (397.73) | 10.29 | +3.33 (47.74) | 5.48 | +0.37 (7.19) |

are limited when applied to individuals with multiple health conditions. Being constrained by linear assumptions, they may not fully capture the complexities inherent in such populations. Conversely, ML models uncover these complexities by learning the variable interactions and maintaining robustness in high-dimensional settings. They consistently reach closer to the estimated error bounds, indicating their superior predictive capability. Not only the use of ML models, considering a broader set of features (both benchmarks use only 7 core features) is proven to be beneficial for predictive performance. Our analysis also highlights that while some of these 64 variables may not be strongly predictive in isolation, their consideration in a multivariate ML model offers improved predictions for comorbid populations, which are frequently underserved by existing prediction guidelines. Take *triglycerides* as an example, which is ranked 4th in the permutation-based variable ranking for comorbid groups in

diabetes prediction (see Table 3), whereas its individual AUC-ROC score is only 0.48. Similarly, while HbA1c is a strong standalone predictor, its predictive capacity is further enhanced when combined with other features, demonstrating that a combination of features is essential to achieving the best-performing model. Notably, a model with 25 predictors achieves slightly better performance than the all-variable model. In this reduced set, ELFs exhibit greater relative importance, underscoring their contribution to risk prediction for comorbid populations. Secondly, leveraging global and local model-agnostic interpretation approaches on our ML models, which incorporate ELFs and proximal risk factors, such as physical measures and activity, diet, and substance use, is instrumental in discovering potentially novel risk predictors for CVD and diabetes. This approach enlarges the set of variables traditionally considered relevant for predicting CVD and diabetes outcomes, thereby advancing our understanding of multimorbidity by revealing complex interactions among these variables, including both predictive and explanatory features. ELFs appear as key drivers of both performance and prediction when combined with proximal factors in comorbid groups, underscoring their role in improving the accuracy of CVD and diabetes risk predictions and understanding complex disease interactions. For instance, being *breastfed as a baby* for CVD, and *having someone to take to the doctor when needed as a child* for diabetes demonstrates information value in comorbid scenarios. While acknowledging the information value of these factors, it is crucial to recognize that breastfeeding is influenced by broader socioeconomic, educational, and cultural contexts [42], and having accessible medical care in childhood is a potential marker of physical neglect [43]. These elements highlight the complexity of isolating direct impacts on health outcomes, emphasizing the need for careful interpretation when such factors appear significant in predictive models for specific individuals. Our results support the claim that ELFs contribute to risk prediction in comorbid groups. While our study provides an initial understanding of their role in multimorbid populations, their contributions may vary within broader populations and, thus, further research is needed to comprehend how these findings generalize to more general populations. Additionally, this study has identified the crucial role of metabolic health measures, including both non-laboratory and laboratory variables, as well as psychological factors, in enhancing the accuracy of predicting CVD and diabetes for comorbid groups.

## Clinical implications

Our findings suggest that ML models may offer an improvement over traditional risk assessment tools such as the Framingham score and DiabetesUK, a gain that is particularly relevant for patients with comorbid conditions. The ML models appear to capture comorbid-specific risk factors, including influences from early life, that are not usually considered by existing algorithms. This indicates the potential for utilizing ML models in clinical settings for more personalized risk assessments since such models are robust in handling high-dimensional data, which facilitates the incorporation of life-course factors into risk models. The use of *post-hoc* interpretative methods on these models aids in making complex data more understandable, which could support clinicians in making informed decisions. However, further validation and research are necessary to fully grasp the implications of these findings for clinical practice, especially in the management and prevention of CVD and diabetes among patients with comorbidities. It is important to note that the 64-variable model, as used in this study, is not intended for direct clinical use. Rather, it serves as a tool to explore the role of ELFs as both performance drivers and prediction drivers in risk models for comorbid patients. Insights from this analysis aim to raise awareness of the importance of ELFs in risk

prediction, encouraging the integration of key predictors, including ELFs, into future clinical models.

## Limitations

The main limitation of our study is that the exclusive use of the UKB affects the generalizability of our findings. With over 96% of participants being white and the age range of participants limited to 40–70 years at the time of assessment, the ethnic and age diversity is limited. Additionally, most of the non-laboratory variables in the UKB, such as health ratings, are obtained through an automated touchscreen questionnaire, a practice that may introduce self-reporting biases, potentially affecting the performance of the models. Furthermore, our study's scope for investigating the contribution of early life factors is limited by the inclusion of only eight of these variables. Recent evidence [6,44] suggests that a wider variety of ELFs should be considered, such as perinatal characteristics like birth weight and prematurity, along with parental and neighborhood characteristics, within the context of more proximal risk factors.

Another limitation of our study is the performance of the models for predicting individuals with comorbid conditions at baseline. Addressing this issue could involve exploring alternative ML model families that might uncover different types of relationships within the data, potentially enhancing performance for this group. Crucially, observed differences in the smallest expected error (Bayes error) between comorbid and non-comorbid groups indicate that no model can achieve zero discrimination in performance without additional information. Therefore, integrating a broader spectrum of input variables may further enhance predictive accuracy for individuals with comorbid conditions. We encourage further research aimed at identifying more informative features and fostering greater transparency around the predictors used in these models, which may ultimately improve prediction performance for comorbid groups. We also note that our models identify statistical associations between variables rather than causal pathways. Therefore, utilizing *causal* machine learning would be instrumental in exploring potential causal links between ELFs and multimorbidity outcomes in more diverse population-based samples.

## Supporting information

**S1 Table. Summary of all UKB input variables used in the study.**
(PDF)

**S1 Fig. Flow diagram of participant stratification based on early-life factors (ELFs), detailing CVD and diabetes outcomes in non-comorbid and comorbid subgroups.**
(PDF)

**S2 Table. Permutation-based variable ranking for predicting CVD in non-comorbid and comorbid groups.**
(PDF)

**S2 Fig. Comprehensive overview of participant profiles for the final study cohort with CVD outcomes: (a) Demographic and socioeconomic distribution, illustrating the population sample's diversity.** (b) Proportion of early-life factors, underlining their frequency and potential impact on later CVD health outcomes.
(PDF)

**S3 Table. Permutation-based variable ranking for predicting diabetes in non-comorbid and comorbid groups.**
(PDF)

**S3 Fig. Comprehensive overview of participant profiles for the final study cohort with diabetes outcomes: (a) Demographic and socioeconomic distribution, illustrating the population sample's diversity.** (b) Proportion of early-life factors, underlining their frequency and potential impact on later diabetes health outcomes.
(PDF)

**S4 Table. Differential top features by effect size: comorbid vs. non-comorbid.** The top 30 features are ranked by their effect size for CVD.
(PDF)

**S4 Fig. Predictive ability of individual variables for CVD and diabetes prediction in comorbid and non-comorbid groups.** Complements Fig 3 by showing the full distribution of AUC-ROC values for all features.
(PDF)

**S5 Table. Differential top features by effect size: comorbid vs. non-comorbid.** The top 30 features are ranked by their effect size for diabetes.
(PDF)

**S5 Fig. Shap scores of the top 30 influential variables in CVD prediction model for non-comorbid and comorbid groups.** Variables marked with an asterisk (*) are those currently employed in current risk assessment models.
(PDF)

**S6 Fig. Shap scores of the top 30 influential variables in diabetes prediction model for non-comorbid and comorbid groups.** Variables marked with an asterisk (*) are currently employed in current risk assessment models.
(PDF)

## Author contributions

**Conceptualization:** Vien Ngoc Dang, Jerónimo Hernández-González.

**Data curation:** Vien Ngoc Dang.

**Formal analysis:** Vien Ngoc Dang.

**Funding acquisition:** Karim Lekadir.

**Investigation:** Vien Ngoc Dang.

**Methodology:** Vien Ngoc Dang, Jerónimo Hernández-González.

**Resources:** Karim Lekadir.

**Supervision:** Jerónimo Hernández-González, Karim Lekadir.

**Validation:** Vien Ngoc Dang.

**Visualization:** Vien Ngoc Dang.

**Writing – original draft:** Vien Ngoc Dang, Jerónimo Hernández-González.

**Writing – review & editing:** Vien Ngoc Dang, Charlotte Cecil, Carmine M. Pariante, Jerónimo Hernández-González.

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
