## [Decision Letter · Decision Letter 0]

7 Apr 2025

PDIG-D-25-00125Characterizing the role of early life factors in machine learning-based multimorbidity risk predictionPLOS Digital Health Dear Dr. Dang, Thank you for submitting your manuscript to PLOS Digital Health. After careful consideration, we feel that it has merit but does not fully meet PLOS Digital Health's publication criteria as it currently stands. Therefore, we invite you to submit a revised version of the manuscript that addresses the points raised during the review process. Please submit your revised manuscript within 60 days Jun 06 2025 11:59PM. If you will need more time than this to complete your revisions, please reply to this message or contact the journal office at digitalhealth@plos.org. Please include the following items when submitting your revised manuscript:* A rebuttal letter that responds to each point raised by the editor and reviewer(s). You should upload this letter as a separate file labeled 'Response to Reviewers'. This file does not need to include responses to any formatting updates and technical items listed in the 'Journal Requirements' section below.* A marked-up copy of your manuscript that highlights changes made to the original version. You should upload this as a separate file labeled 'Revised Manuscript with Track Changes'.* An unmarked version of your revised paper without tracked changes. You should upload this as a separate file labeled 'Manuscript'. If you would like to make changes to your financial disclosure, competing interests statement, or data availability statement, please make these updates within the submission form at the time of resubmission. Guidelines for resubmitting your figure files are available below the reviewer comments at the end of this letter. We look forward to receiving your revised manuscript. Kind regards, Gloria Hyunjung KwakSection EditorPLOS Digital Health Gloria Hyunjung KwakSection EditorPLOS Digital Health Leo Anthony CeliEditor-in-ChiefPLOS Digital Healthorcid.org/0000-0001-6712-6626 **Journal Requirements:**

1. We ask that a manuscript source file is provided at Revision. Please upload your manuscript file as a .doc, .docx, .rtf or .tex. **Additional Editor Comments (if provided):****Reviewers' Comments:** Reviewer's Responses to Questions

**Comments to the Author**

1. Does this manuscript meet PLOS Digital Health’s publication criteria? Is the manuscript technically sound, and do the data support the conclusions? The manuscript must describe methodologically and ethically rigorous research with conclusions that are appropriately drawn based on the data presented.

Reviewer #1: Yes

Reviewer #2: Yes

2. Has the statistical analysis been performed appropriately and rigorously?

Reviewer #1: Yes

Reviewer #2: Yes

3. Have the authors made all data underlying the findings in their manuscript fully available (please refer to the Data Availability Statement at the start of the manuscript PDF file)?

Reviewer #1: Yes

Reviewer #2: Yes

4. Is the manuscript presented in an intelligible fashion and written in standard English?

Reviewer #1: Yes

Reviewer #2: Yes

5. Review Comments to the Author

Reviewer #1: Dear Author,

This study analyzes the contribution of ELFs to machine-learning-based risk prediction models for comorbid populations. Yo also utilized incorporation of life-course factors into risk models. Integrating a diverse range of physiological, psychological, and ELFs. I anm wobdering about integrsation of such a multilayered data to develop an AI moder with a highly heterogenous data??. The abstract is unstructured.

Reviewer #2: This paper uses a series of machine learning models and analyses to

attempt to characterize the role of early life factors in

multimorbidity risk prediction. Specifically, the authors hope to use

an examination the role of these factors in predictive models to

inform an understanding of how they might impact outcomes.

The paper is through and interesting, including a broad range of

analyses that provide substantial understanding. Although there are a

number of points that could use clarification, I did not identify any

major methodological concerns.

I am curious about one important point that was not addressed in the

paper. If the goal was to address the role of early life factors, why

start with machine learning models as the preferred analysis approach?

Causal methods, such as propensity scoring or causal structural

discovery, might yield some important insights, particularly

regarding potential interactions between observed factors. The reasons

for the selected approaches should be given and contrasted with

alternative means of addressing questions of interest.

My other major concern with this paper is that the number of analyses

is large and the volume of data is somewhat overwhelming. Some

additional guidance, perhaps in terms of a workflow figure detailing

the types of analyses and the goals presented might make the paper

easier to read.

A number of minor readability/comprehensibility issues arose during my

reading of the paper:

1. In terms of selection of valuables (line 137-139), it might help to

update Table S1 to indicate which factors are early life factors.

2. Same lines, how are the 64 variables processed into 83 features?

Details should be provided.

3. Value of information - 207-217. Particularly since "value of

information" has a specific meaning in many contexts, this term

should be defined in more detail.

4. Same section - the group-specific Bayes error estimation methods

might not be familiar to all readers and should be described in a bit

more detail.

5. I don't understand what is meant by the comment that "metrics like

SHAP explain how variables contribute to performance but do not

directly quantify their contribution to predictive performance. I

assume that the point here is that SHAP is simply the result of

multiple models that approximate the original model, but I am not quite

sure. This somewhat cryptic statement should be replaced with

additional clarification.

6. The description of "Group-specific Bayes errors" is confusing,

mentioning noise estimates that were not discussed when the

Group-specific errors where introduced earlier. Although it's clear on

a second read that "noise estimates" and "errors" are meant to be

interchangeable, this paragraph would benefit from some clarification.

7. Figure 1 - the caption needs to be edited. "Framming score" is not

correct.

8. Figure 2 - the graphs as presented are a bit hard to read due to

the difference in the range covered in the x- and y- axes. For

example, the CVD graph seems to range from roughly 0.4- 0.6 on the

x-axis and 0.45-0.63 on the y axis. This difference leads the

equivalence (y=x) line to be skewed to the right, thus complicating

the interpretation of which was better. A parallel presentation with

the y=x line being a straight line from lower-left to upper right,

would be much clearer.

9. The discussion of HbA1c's role in diabetes prediction reads more

like methods than results. Perhaps this should be moved to the methods

section.

10. Finally, there are a great many results in this paper that are

hard to interpret. Some sort of tabular summary and/or clearer

delineation of the topics in the discussion would make the paper more

readable and the conclusions more digestible.

6. PLOS authors have the option to publish the peer review history of their article (what does this mean?). If published, this will include your full peer review and any attached files.

**Do you want your identity to be public for this peer review?** For information about this choice, including consent withdrawal, please see our Privacy Policy.

Reviewer #1: No

Reviewer #2: **Yes: **Harry Hochheiser

---

## [Decision Letter · Decision Letter 1]

29 Jun 2025

PDIG-D-25-00125R1Characterizing the role of early life factors in machine learning-based multimorbidity risk predictionPLOS Digital Health Dear Dr. Dang, Thank you for submitting your manuscript to PLOS Digital Health. After careful consideration, we feel that it has merit but does not fully meet PLOS Digital Health's publication criteria as it currently stands. Therefore, we invite you to submit a revised version of the manuscript that addresses the points raised during the review process. Please submit your revised manuscript within 30 days Jul 29 2025 11:59PM. If you will need more time than this to complete your revisions, please reply to this message or contact the journal office at digitalhealth@plos.org. Please include the following items when submitting your revised manuscript:* A rebuttal letter that responds to each point raised by the editor and reviewer(s). You should upload this letter as a separate file labeled 'Response to Reviewers'. This file does not need to include responses to any formatting updates and technical items listed in the 'Journal Requirements' section below.* A marked-up copy of your manuscript that highlights changes made to the original version. You should upload this as a separate file labeled 'Revised Manuscript with Track Changes'.* An unmarked version of your revised paper without tracked changes. You should upload this as a separate file labeled 'Manuscript'. If you would like to make changes to your financial disclosure, competing interests statement, or data availability statement, please make these updates within the submission form at the time of resubmission. Guidelines for resubmitting your figure files are available below the reviewer comments at the end of this letter. We look forward to receiving your revised manuscript. Kind regards, Gloria Hyunjung KwakSection EditorPLOS Digital Health Gloria Hyunjung KwakSection EditorPLOS Digital Health Leo Anthony CeliEditor-in-ChiefPLOS Digital Healthorcid.org/0000-0001-6712-6626**Reviewers' Comments:** Reviewer's Responses to Questions

**Comments to the Author**

1. If the authors have adequately addressed your comments raised in a previous round of review and you feel that this manuscript is now acceptable for publication, you may indicate that here to bypass the “Comments to the Author” section, enter your conflict of interest statement in the “Confidential to Editor” section, and submit your "Accept" recommendation.

Reviewer #2: (No Response)

Reviewer #3: (No Response)

2. Does this manuscript meet PLOS Digital Health’s publication criteria? Is the manuscript technically sound, and do the data support the conclusions? The manuscript must describe methodologically and ethically rigorous research with conclusions that are appropriately drawn based on the data presented.

Reviewer #2: Yes

Reviewer #3: Yes

3. Has the statistical analysis been performed appropriately and rigorously?

Reviewer #2: Yes

Reviewer #3: I don't know

4. Have the authors made all data underlying the findings in their manuscript fully available (please refer to the Data Availability Statement at the start of the manuscript PDF file)?

Reviewer #2: Yes

Reviewer #3: Yes

5. Is the manuscript presented in an intelligible fashion and written in standard English?

Reviewer #2: Yes

Reviewer #3: Yes

6. Review Comments to the Author

Reviewer #2: This revised paper addresses most of my concerns. Thanks to the

authors for their attention to comments. I have two outstanding

concerns:

1 I appreciate your change in language regarding "Value of

information", but I find the revision to be inadequate. "Value of

information" is an accepted phrase in the literature, involving

structured methods for determining the marginal value of information

in decision-making. See, for example, Heath, et al. 2017 (DOI

10.1177/0272989X176976) for an overview. As this paper does not use

any of those methods, calling this section a VOI analysis is

inappropriate. I suggest rephrasing this section, perhaps labeling

it as "analysis of contributions of predictive factors".

2. I appreciate the response regarding the graphs comparing the ROCs

for comorbid and non-comorbid groups, but I believe the groups are

still misleading. I suggest revising both graphs to have 0.5 as the

lowest point for both the x- and y-axes. This would provide the

desired 45 degree angle while avoiding both the compression that might

be associated with showing the whole range and the difficulties of

comparison created by having the range of values on the axes differ

for the two graphs.

Two minor points

1. Please provide version numbers for libraries mentioned in the

methods section.

2. Consider making the code available as open source.

Reviewer #3: The objective of this study is to study how machine learning can investigate the role of early-life factors in predicting cardiovascular disease and diabetes. This essay is an attempt to contribute to the predictive model literature by showing how to represent and integrate early-life factors in model development and of the importance of doing so. This is challenging due to the complexity of the relevant variables, their nonlinear interactions, and the assumptions a model might make. The hope is to provide insights that enhance the accuracy of existing risk prediction algorithms, specifically tailored for individuals with comorbid conditions.

I found the essay intriguing, very well written, and admirably detailed in its methodological description and findings. Quite frankly, this is a “big” article. I’m not trained in the data science dimensions that inform the study design so I must defer to other reviewers who can comment on their validity. I can say, though, that the effort to study early-life factors seems to me quite valuable, such that even if readers find material to criticize, the issues presented in this paper should be of value to the digital health community. Nevertheless, it would seem to me that the only persons with enough background to critique the methodology of the study would be data science specialists. Consequently, their review of this study is of quintessential importance.

Three drawbacks of the study are already mentioned by the authors. The first, that the model was normed using a United Kingdom database, with over 96% of participants being white. Another problem might be the accuracy of the data being entered, especially as it reports on early life factors. One is especially worried about data that are not only inaccurate but missing. A third factor is simply the plethora of variables programmed into the model. I suppose one can question whether that degree of detail is necessary or really contributes to the model’s predictive accuracy—although that question would obviously be addressed in future test applications of the model.

Ultimately, I found this essay very impressive in virtually every way. Should other reviewers find the methodology and findings acceptable, I would strongly endorse the article’s acceptance and eventual publication.

7. PLOS authors have the option to publish the peer review history of their article (what does this mean?). If published, this will include your full peer review and any attached files.

**Do you want your identity to be public for this peer review?** For information about this choice, including consent withdrawal, please see our Privacy Policy.

Reviewer #2: **Yes: **Harry Hochheiser

Reviewer #3: No

---

## [Editor Report · Decision Letter 2]

23 Jul 2025

Characterizing the role of early life factors in machine learning-based multimorbidity risk prediction

PDIG-D-25-00125R2

Dear Dr. Dang,

We're pleased to inform you that your manuscript has been judged scientifically suitable for publication and will be formally accepted for publication once it meets all outstanding technical requirements.

Within one week, you'll receive an e-mail detailing the required amendments. When these have been addressed, you'll receive a formal acceptance letter and your manuscript will be scheduled for publication.

An invoice for payment will follow shortly after the formal acceptance. To ensure an efficient process, please log into Editorial Manager at https://www.editorialmanager.com/pdig/ click the 'Update My Information' link at the top of the page, and double check that your user information is up-to-date. For billing related questions, please contact billing support at https://plos.my.site.com/s/.

Kind regards,

Sarah Mayo

Staff Admin

PLOS Digital Health